Impact of a multicomponent physical exercise program on intrinsic capacity in community-dwelling older adults

Felipe Sarah Giulia 1
Printes Clarissa Biehl 1 2
http://orcid.org/0000-0002-7695-6020 Sato Douglas Kazutoshi 1
http://orcid.org/0000-0003-1937-6393 Baptista Rafael Reimann 1 2 3 rafael.baptista@pucrs.br
1 Pontifícia Universidade Católica do Rio Grande do Sul , Porto Alegre , Brazil
2 Institute of Geriatrics and Gerontology , Porto Alegre, RS , Brazil
3 BioHub Health Innovation Hub , Porto Alegre, RS , Brazil
Khoo Selina
Electronic publication date: 2025 Mar 12
Publication date: 2025
Volume: 13
Electronic Location ID: e19017
Received 2024 Nov 1; Accepted 2025 Jan 28
Copyright: © 2025 Felipe et al.
Copyright year: 2025
Copyright holder: Felipe et al.
License: This is an open access article distributed under the terms of the Creative Commons Attribution License, which permits unrestricted use, distribution, reproduction and adaptation in any medium and for any purpose provided that it is properly attributed. For attribution, the original author(s), title, publication source (PeerJ) and either DOI or URL of the article must be cited.
License URL: https://creativecommons.org/licenses/by/4.0/

Keywords: Exercise, Older adults, Intrinsic capacity

Funding: The Coordination for the Improvement of Higher Education Personnel (CAPES) Sarah Giulia Bandeira Felipe The Coordination for the Improvement of Higher Education Personnel (CAPES) provided a full scholarship to Sarah Giulia Bandeira Felipe. The funders had no role in study design, data collection and analysis, decision to publish, or preparation of the manuscript.

==============================
Introduction

In 2015, the World Health Organization introduced intrinsic capacity, a health indicator encompassing five domains aimed at promoting healthy aging. Multicomponent exercise programs are recommended to maintain and optimize intrinsic capacity; however, evidence on their effects in robust older adults is limited. This study aimed to evaluate the effects of a multicomponent exercise program on intrinsic capacity in older adults.

Methods

Older adults (≥60 years) enrolled in a multicomponent training program in Porto Alegre (RS) were evaluated for intrinsic capacity using specific tests for each domain. The total score, ranging from 0 to 10 points, was obtained by summing the scores of the five domains. Participants underwent a 12-week multicomponent exercise intervention and were reassessed.

Results

After the 12-week intervention, the composite intrinsic capacity score significantly increased. Significant improvements were found in the cognitive, vitality and locomotion domains. The practice of Chinese curative gymnastics contributed to a one-point increase in the composite intrinsic capacity score. No association was found between frequency, duration of training, or the number of exercise modalities and the composite intrinsic capacity score.

Conclusion

The multicomponent exercise program positively influenced composite intrinsic capacity, with notable improvements in vitality, cognition, and mobility. Personalization and individualization of training, combined with health education and social integration, were crucial in promoting healthy aging in the studied sample.

Introduction

In 2015, the World Health Organization (WHO), in its World Report on Ageing and Health, introduced two important concepts: intrinsic capacity and functional ability. Functional ability relates to the health attributes that enable people to be or do what is important to them, combining intrinsic capacity, the environment, and the interaction of individuals with their environment. Intrinsic capacity, on the other hand, refers to all the physical and mental capacities a person can utilize (World Health Organization, 2015).

Building on these two concepts, healthy aging is defined as the process of developing and maintaining the functional ability that enables well-being in older age. Currently, one of the global challenges for rapidly aging societies is to promote healthy aging, regardless of health conditions or the presence of diseases. The main recommendations focus on integrated, person-centered care models that replace disease-based curative models (World Health Organization, 2020).

One way to achieve this is by building and maintaining intrinsic capacity over the years, as this concept is related to the continuous measurement of the capabilities of multiple biological systems (rather than limitations) most relevant to healthy aging. Given the dynamic nature of this construct over the lifespan, monitoring its trajectory offers valuable insights for clinical practice and public health at both individual and population levels (George et al., 2021).

Operationally, intrinsic capacity can be understood through five domains: locomotion, vitality, sensory (vision and hearing), cognition, and psychological. When evaluated together, these domains integrate into the composite intrinsic capacity. Previous studies have demonstrated the impairment of intrinsic capacity in older adults (Stolz et al., 2022), (Leung et al., 2022; Zhang et al., 2023), and have also predicted adverse health outcomes such as falls, frailty, and disability (Yu et al., 2021; Locquet et al., 2022; Zhou & Ma, 2022).

A scoping review highlighted that a lower intrinsic capacity score is associated with a 7% increase in the risk of dependency for basic activities of daily living (BADL). Additionally, poor quality of life contributes to the decline in intrinsic capacity, with a one-point decrease in intrinsic capacity linked to a 6% increase in the risk of long-term care admissions. Conversely, higher intrinsic capacity scores are associated with a reduced risk of frailty and falls (Zhou, Kuang & Hu, 2023).

The aforementioned data highlights the necessity for early interventions to manage the decline in intrinsic capacity and the occurrence of adverse health events. In response, the WHO published the Integrated Care for Older People (ICOPE) manual in 2019, which outlines strategies to optimize intrinsic capacity within primary care settings. Among the recommended interventions, regular physical exercise is highly emphasized across various domains (World Health Organization, 2019).

Multicomponent physical exercise programs, which combine various modalities (strength training, aerobic exercise, flexibility, and balance), are more effective in maintaining functionality and preventing disabilities. Additionally, these programs are crucial for preserving mobility, musculoskeletal function, and the optimal functioning of other body systems (neurological, cardiovascular, respiratory, digestive, endocrine) that impact intrinsic capacity (Izquierdo, 2018).

Another benefit highlighted in the literature is social engagement and psychological well-being. Exercise also acts as a protective factor in managing and preventing chronic non-communicable diseases and various other health conditions, such as frailty and falls (De Maio et al., 2022). Guidelines for older adults indicate that any type of exercise provides health improvements, with the intensity level matching individual capacity (Nelson et al., 2007).

In Japan, a clinical trial demonstrated that both aerobic and strength training benefited intrinsic capacity among older adults (Huang et al., 2021). Similarly, a study in Spain found that a 12-week multicomponent exercise program improved intrinsic capacity, particularly in the domains of locomotion, cognition, and vitality among community-dwelling older adult (Sánchez-Sánchez et al., 2022).

While these studies show effects on individual domains of intrinsic capacity and on composite intrinsic capacity (the combination of all domains), they were conducted with older adults with subjective memory problems (Huang et al., 2021), mild cognitive impairment, and pre-frailty/frailty (Sánchez-Sánchez et al., 2022). Thus, generalizing these findings to the broader population is challenging.

Therefore, evidence remains limited regarding the application of this intervention across all domains of intrinsic capacity, especially in Brazil. To our knowledge, this will be the first study conducted in the country with community-dwelling older adults who are apparently robust and not in a state of pre-frailty or cognitive impairment.

The literature also indicates that lifestyle behaviors, including physical exercise, are strongly associated with the maintenance of intrinsic capacity (Muneera, Muhammad & Althaf, 2022; Zhou, Kuang & Hu, 2023; Arias-Casais et al., 2022). Additionally, there is a scarcity of studies addressing changes in intrinsic capacity over time. Currently, no global index of intrinsic capacity is validated for clinical practice and research, underscoring the need for more research on how to measure and intervene in intrinsic capacity and its domains in different settings (Gonzalez-Bautista et al., 2020). Therefore, the aim of this study was to evaluate the effects of a multicomponent exercise program on intrinsic capacity among older adults.

Materials and Methods

This study was approved by the Research Ethics Committee of the Pontifical Catholic University of Rio Grande do Sul (PUCRS) under approval number 5.517.315/CAAE: 60234322.1.0000.5336, adhering to the regulations outlined in Resolution 466/12 of the National Health Council for research involving human subjects, with the respective signing of the Informed Consent Form (ICF).

This research is a 12-week, quasi-experimental, time-series study, prospective in nature, with initial and final evaluations. The sample was non-probabilistic, calculated based on a previous study by Sánchez-Sánchez et al. (2022), resulting in a sample size of 31 older adults, including anticipated dropouts.

Recruitment occurred through a physical exercise program in community located in X, via phone calls and messages, social media announcements, and the local team. All older adults enrolled between July and September 2023 were invited to participate in the study. The detailed flowchart can be seen in Fig. 1.

Figure 1 Flowchart.

Flowchart of the study stages and assessments.

We established the inclusion criteria as follows: individuals aged 60 years or older; having pre-intervention and post-intervention assessments of all five domains (cognition, vitality, psychological, locomotor, and sensory); attending the program at least twice a week (24 sessions in 12 weeks) and participating in two modalities for 90 min per day. Exclusion criteria included: physical and cognitive impairments that prevent or limit assessments; sequelae of stroke; Parkinson’s and Alzheimer’s diseases; recent surgeries within the last 12 months (hand, hip, or knee); and absence from the program for more than 15 days.

Pre-intervention data collection occurred in September 2023 and post-intervention in December 2023. During this period, sociodemographic, clinical, and social data were collected, and the following scales were applied: Geriatric Depression Scale—GDS-15 (Yesavage et al., 1982), International Physical Activity Questionnaire (IPAQ) (Marshall & Bauman, 2021; Craig et al., 2003), Short Form 6 dimensions (SF-6D Brazil) Quality of Life Questionnaire (Brazier et al., 1998), Functional Independence Measure (Rikli & Jones, 1999), and Montreal Cognitive Assessment (MoCA) (Nasreddine et al., 2005; Freitas et al., 2013). Self-reported assessments of the sensory domain were conducted based on specific questions about vision and hearing, extracted from the ICOPE protocol. The questions included: “Do you wear glasses?” and “Do you have difficulty seeing far or near?” for vision, and “Do you use a hearing aid?” and “Do you have difficulty hearing whispers?” for hearing (World Health Organization, 2020).

Additionally, we assessed weight (kg), height (cm), body mass index (BMI, kg/m2), current body fat percentage (%), muscle mass percentage (%), visceral fat percentage (%), basal metabolic rate (kcal), body age (years), handgrip strength (kgf), lower limb strength (number of repetitions), functional mobility (seconds), abdominal circumference (cm), waist circumference (cm), thigh circumference (cm), calf circumference (cm), systolic and diastolic blood pressure (mmHg), heart rate (bpm), flexibility (cm), and balance (seconds). Details of this evaluation can be found at: https://doi.org/10.7910/DVN/UI5SGL.

To select the variables for measuring each domain of composite intrinsic capacity, we reviewed scientific articles on the topic and subsequently held a research group discussion to identify the most relevant variables available. For the final scoring, we chose to use the scoring proposed in the study by López-Ortiz et al. (2022), where composite intrinsic capacity can range from zero (worst intrinsic capacity) to ten (best intrinsic capacity). The domains of intrinsic capacity were evaluated as shown in the Table 1.

Table 1 Assessment of intrinsic capacity domains.

Intrinsic capacity	Category	Instrument	
Vitality	Handgrip strength	Dynamometry	
Body mass index	Scale	
Weight loss and lack of appetite	-Have you unintentionally lost ≥3 kg in the last 3 months? How many kg?
-Have you experienced a loss of appetite?	
Sensory	Vision and hearing	Self-report questions:
-Do you wear glasses? Are you nearsighted or farsighted?
-Do you use a hearing aid? Do you have difficulty hearing whispers?	
Locomotion	Functional mobility	Timed Up and Go Test	
Leg strength
Flexibility
Balance	Sit-to-stand test
Sitting and Reaching Test
Unipedal Balance Test	
Psychological	Depressive symptoms	Geriatric Depression Scale	
Cognitive	Global cognitive screening	Montreal Cognitive Assessment	

In the locomotion domain, for the Timed Up and Go test (Podsiadlo & Richardson, 1991), a value of 0.5 points was assigned for times under 10 s, and a value of 0 was assigned for times over 10 s. For the sit-to-stand test (Rikli & Jones, 1999, 2013), a value of 0 was assigned for the number of repetitions below the age- and sex-specific threshold, and a value of 0.5 was assigned for the number of repetitions equal to or above the threshold. In the sit-and-reach test, a score of 0.5 was given for flexibility within the recommended range for age and sex, and a score of 0 for flexibility outside the recommended range. Finally, in the one-leg balance test, values of 20 s or less received a score of 0, and values of 21 s or more received a score of 0.5. The locomotion score was obtained by summing the values from the Timed Up and Go test, sit-to-stand test, sit-and-reach test, and one-leg balance test, with the final score ranging from 0 to 2.

In the vitality domain, for BMI, values below 22 kg/m2 and above 27 kg/m2 received a score of 0, while values between 22–27 kg/m2 received a score of 0.5 (Lipschitz, 1994). For handgrip strength (HGS), a score of 0 was assigned for values below 27 kgf for men and below 16 kgf for women. A score of 0.5 was assigned for values equal to or above 27 kgf for men and 16 kgf for women (Cruz-Jentoft et al., 2019). Weight loss was scored as follows: 0 for weight loss of 3 kg or more in the past 3 months and 0.5 for weight loss of less than 3 kg or no weight loss. For the variable appetite loss, a score of 0 was assigned for “yes” responses and 0.5 for “no” responses. The vitality score was obtained by summing the values for BMI, HGS, weight loss, and appetite loss, with the final score ranging from 0 to 2.

In the cognition domain, following the administration of the MoCA test (Nasreddine et al., 2005; Freitas et al., 2013), a score of 0 was assigned for results below the cut-off point according to educational level, and a score of 2 for results above the cut-off point. Thus, the final score for the cognition domain ranged from 0 to 2.

For the psychological domain, based on the GDS-15 scale (Yesavage et al., 1982), a score of 0 was assigned for values between 11 and 15; an intermediate score of 1 for values between 6 and 10; and the maximum score of two points for values between 0 and 5. Therefore, the final score for the psychological domain ranged from 0 to 2.

In the sensory domain, self-reported questions were used to establish scores for vision and hearing (World Health Organization, 2019). For vision, a score of 0 was assigned to those who have difficulty seeing far or near and wear prescription glasses; the highest score of 1 was assigned to those who have no difficulty seeing far or near and do not wear prescription glasses. For hearing, a score of 0 was assigned to those who have difficulty hearing whispers and do not use hearing aids; the highest score of 1 was assigned to those who have no difficulty hearing whispers and do not use hearing aids.

The final score for the sensory domain was obtained by summing the scores for vision and hearing, with the total score ranging from 0 to 2.

The composite intrinsic capacity score will be calculated by summing the scores of each domain, totaling 10 points.

CI = locomotor (2 points) + vitality (2 points) + cognition (2 points) + psychological (2 points) + sensorial (2 points).

Total = 10 points.

Following this stage, the exercise prescription was given, and the participants began the multicomponent training.

The 12-week multicomponent training program included individual and group sessions (10–20 people), lasting 45 min, five times per week. The exercise prescription was conducted according to the guidelines of the American College of Sports Medicine (Nelson et al., 2007) and the Ministry of Health (Brazil, 2021). The program included the following modalities: weight training, aerobic exercises, circular dance, Chinese therapeutic gymnastics, Pilates matwork, gerontomotricity, pedestrianism, maintenance gymnastics, change games, aquatic activities, and Nordic walking.

The selection of these modalities was conducted by a qualified training team in a personalized manner, respecting individual needs based on a prior physical assessment. Strength training and cardiovascular training were recommended for at least two sessions per week, while other activities could be performed up to three times per week. During the intervention period, the training team remained the same, and for most modalities, the sessions were led by a specific trainer. Additionally, participant preferences were taken into account to maximize engagement in the program

Adherence monitoring was carried out using a specific log, where older adults recorded the date of each training session. After the 13th training session, the exercise intensity and volume parameters were adjusted. Progression was established with a 5% increase in the total load, accompanied by a reduction in the number of repetitions, which decreased from 15 to 10. The detailed recording of the sessions enabled rigorous monitoring of participant regularity and commitment to the program, contributing to adherence and effective progression management. Detailed information on the modalities and adjustments made is provided in the Supplemental Material (https://doi.org/10.7910/DVN/UI5SGL).

During the program, participants also received health education through lectures by specialists. Topics covered included chronic disease management, healthy eating, reducing sedentary behavior, and medication management, among others.

Twelve weeks after the intervention, all participants were re-evaluated and questioned about the modalities practiced, their frequency, and the duration of each session. The study flowchart can be viewed in Fig. 2.

Figure 2 Flowchart.

It is important to note that this study did not randomly allocate participants into intervention and control groups. Instead, the participants themselves chose the modality, frequency, and duration of the activities offered by the program. Additionally, although the modalities had application and conduction protocols, there was inevitably heterogeneity in practice as they were offered by different instructors and attended by different groups of participants. Thus, we aimed to capture the actual participation patterns of older adults in community activities, as a fully controlled study may not necessarily reflect this reality.

Data analysis was performed using SPSS version 28.0 and an intention-to-treat approach was used. Quantitative variables were described by mean and standard deviation or median and interquartile range. Categorical variables were described by absolute and relative frequencies. Comparison between the two time points was evaluated using the Generalized Estimating Equations (GEE) model. The linear model was applied for variables with normal distribution, the logarithmic transformation model was used for variables with asymmetric distribution, the ordinal logistic model was applied for ordinal variables, and the binary logistic model was used for dichotomous variables.

Associations between the frequency and duration of training with changes in intrinsic capacity domain scores were evaluated using Spearman’s correlation coefficient. The comparison of changes in intrinsic capacity domain scores according to the training modalities was performed using the Mann-Whitney test. The significance level adopted was 5% (p < 0.05). The raw data are available at: https://doi.org/10.5281/zenodo.14002534.

Results

We evaluated 43 participants in this study, with a mean age of 67.7 years (standard deviation ± 4.3). Women comprised 86.0% of the sample. The majority of participants had completed high school and higher education (23.3%), were married or had a partner (44.2%), were retired (86.0%), and lived alone (39.5%). The average number of children was 2.0 (standard deviation ± 1.2), and most reported having a support network (93.0%); however, only 20.9% mentioned attending community centers for the older adults.

At baseline, most participants had some comorbidity (90.7%), particularly hypertension (48.8%), dyslipidemia (46.5%), diabetes (25.6%), hypothyroidism (20.9%), and osteoporosis (18.6%). Regarding medication use, 88.4% reported continuous use, with an average of 3.56 medications (standard deviation ± 2.65), the most common being antihypertensives (48.8%), antilipemics (39.5%), hormone replacement therapy (27.9%), hypoglycemics (25.6%), and vitamin supplements (23.3%). Concerning falls, 76.7% of the sample reported not having fallen in the past 12 months.

In terms of lifestyle, 62.8% did not consume alcohol at least once a week, 97.7% did not smoke, 51.2% did not engage in physical exercise, 67.4% did not suffer from insomnia, and 86.0% reported having restorative sleep, averaging 7.12 h per day (standard deviation ± 1.1). Sun exposure was confirmed by 83.7% of the participants.

Regarding the multicomponent training, more than half of the older adults attended the program three times a week (55.0%) and engaged in 90 min of physical activity per attended day (62.5%). The most practiced exercise modalities among the older adults were Chinese therapeutic gymnastics (87.5%), followed by gerontomotricity (80.0%) and maintenance gymnastics (70.0%).

After the 12-week intervention, there was an increase in the composite intrinsic capacity score, with an initial average of 6.9 points (standard deviation ± 1.3), rising to 7.7 points (standard deviation ± 1.9), resulting in a statistically significant difference (p < 0.015). Significant improvements were observed in individual domain analyses of the intrinsic capacity. In the cognitive domain, the score improved from 1.44 ± 1.9 to 1.95 ± 0.3 (p < 0.001); in the vitality domain, from 1.57 ± 0.3 to 1.65 ± 0.3 (p < 0.045); and in the locomotor domain, from 1.38 ± 0.4 to 1.85 ± 0.2 (p < 0.001). No significant improvements were found in the sensory and psychological domains. The results related to the composite intrinsic capacity score and individual domains after the intervention are shown in Table 2.

Table 2 Intrinsic capacity composite score and domain-specific scores pre-intervention and post-intervention.

Variables	Pre-intervention	Post-intervention	p	
Median (P25–P75)/Mean (±)	Median (P25–P75)/Mean (±)	
Score SD	1 (0–1)/0.67 ± 0.4	1 (0–1)/0.68 ± 0.5	0.993	
Score PD	2 (2–2)/1.91 ± 0.3	2 (2–2)/2.00 ± 0.0	0.100	
Score CD	2 (0–2)/1.44 ± 0.9	2 (2–2)/1.95 ± 0.3	<0.001	
Score DV	1.5 (1.5–2)/1.57 ± 0.3	1.5 (1.5–2)/1.65 ± 0.3	0.045	
Score LV	1.5 (1.0–1.5)/1.38 ± 0.4	2.0 (1.5–2.0)/1.85 ± 0.2	<0.001	
Score ICC*	6.9 ± 1.3	7.7 ± 1.9	0.015	
Note:

Values are presented according to the median at the 25th and 75th percentiles and according to the mean and standard deviation (±). SD, sensory domain; PD, psychological domain; CD, cognitive domain; VD, vitality domain; LD, locomotion domain; *ICC, intrinsic capacity composite.

In the individual analysis of the domains, we did not find significant changes in the sensory domain (Table 3).

Table 3 Pre-intervention and post-intervention analysis of variables in the sensory domain.

Variables	Pre-intervention	Post-intervention	p	
n (%)	n (%)	
Glasses	38 (88.4)	38 (95.0)	0.114	
Vision	37 (86.0)	38 (95.0)	0.139	
Hearing aid	2 (4.7)	1 (2.5)	0.364	
Hearing	13 (30.2)	14 (35.0)	0.390	
Note:

Values are presented according to frequency and proportion (%).

In the psychological domain, as measured by the 15-item Geriatric Depression Scale (GDS-15), a reduction in the post-test score was observed, with an initial average of 1.98 points (standard deviation ± 2.2) decreasing to 1.15 points (standard deviation ± 1.5), resulting in a statistically significant difference (p = 0.007) (Table 4).

Table 4 Pre-intervention and post- intervention analysis of variables in the psychological domain.

Variables	Pre-intervention	Post-intervention	p	
Median (P25–P75)/mean (±)	Median (P25–P75)/mean (±)	
Score EDG-15	1 (0–3)/1.98 ± 2.2	1 (0–2)/1.15 ± 1.5	0.007	
Note:

Values presented according to the median at the 25th and 75th percentiles, and according.

In the cognitive domain, significant changes were also found in the MoCA test score, with an initial average of 21.7 points (standard deviation ± 3.3), which increased to 24.1 points (standard deviation ± 2.1) after the intervention (p < 0.001) (Table 5).

Table 5 Pre-intervention and post-intervention analysis of variables in cognitive domain.

Variables	Pre-intervention	Pos-intervention	p	
Mean (±)	Mean (±)	
MOCA test	21.7 ± 3.3	24.1 ± 2.1	<0.001	
Note:

Values presented according to the mean and standard deviation (±). MoCA, Montreal Cognitive Assessment.

Regarding the vitality domain, significant changes were identified only in BMI (27.7 ± 4.7 to 27.2 ± 4.7, p = 0.006) and unintentional weight loss over the past 3 months (from 18.6% to 0.0%, p = 0.002). No significant results were found in the other variables used to measure this domain (Table 6).

Table 6 Pre-intervention and post-intervention analysis of variables in vitality domain.

Variables	Pre-intervention	Pos-intervention	p	
n (%)/mean (±)	n (%)/mean (±)	
BMI (kg/m2)	27.7 ± 4.7	27.2 ± 4.7	0.006	
Handgrip strength	24.9 ± 6.4	24.8 ± 6.0	0.614	
Weight loss	8 (18.6)	0 (0.0)	0.002	
Lack of appetite	3 (7.0)	3 (7.5)	0.914	
Note:

Values presented according to frequency and proportion (%) and mean and standard deviation (±). BMI, Body Mass Index.

In the locomotor domain, significant improvements were observed after the intervention in all tests: sit-to-stand (14.3 ± 4.6 to 17.4 ± 3.7, p < 0.001), Timed Up and Go test (6.87 ± 1.32 to 6.60 ± 1.18, p = 0.030), flexibility (−1.22 ± 8.3 to 4.08 ± 8.4, p < 0.001), and single-leg balance (23.0 ± 9.5 to 28.9 ± 5.0, p < 0.001) (Table 7).

Table 7 Pre-intervention and post-intervention analysis of variables in locomotion domain.

Variables	Pre-intervention	Post-intervention	p	
Mean (±)	Mean (±)	
Sit and stand test	14.3 ± 4.6	17.4 ± 3.7	<0.001	
TUG	6.87 ± 1.32	6.60 ± 1.18	0.030	
Flexibility	−1.22 ± 8.3	4.08 ± 8.4	<0.001	
Balance	23.0 ± 9.5	28.9 ± 5.0	<0.001	
Note:

Values presented according to the mean and standard deviation (±). TUG, Timed Up and Go test.

Furthermore, we found that the composite intrinsic capacity score varied by one point after practicing Chinese therapeutic gymnastics, with a p-value of 0.012, indicating statistical significance.

The combination of strength training and gerontomotricity also significantly modified the composite intrinsic capacity (CIC) score by one point, with a p-value of 0.034. Similar results were obtained with the combination of strength training, gerontomotricity, and Chinese therapeutic gymnastics, with a p-value of 0.034.

The analysis of the association between training frequency and CIC score, and exercise duration and CIC score, revealed a Spearman correlation coefficient of p = 0.037 (p = 0.820) and p = 0.103 (p = 0.528), respectively, indicating a very weak and non-significant correlation between these variables. Regarding the number of modalities practiced and the domain scores, we obtained a statistically significant correlation coefficient (p = 0.37, p = 0.018) in the sensory domain, suggesting that practicing a greater number of modalities influences the score in the sensory domain (Table 8).

Table 8 Association between exercise frequency and duration in changes of scores in intrinsic capacity composite and by domain.

Variables	Frequency	p	Exercise duration	p	Number of modalities		
Spearman’s correlation coefficient	Spearman’s correlation coefficient	Spearman’s correlation coefficiente	p	
Score SD	0.174	0.283	0.273	0.089	0.373	0.018	
Score PD	0.000	1.000	−0.274	0.087	0.147	0.365	
Score CD	0.031	0.847	−0.123	0.451	−0.019	0.907	
Score VD	0.113	0.487	0.088	0.590	−0.137	0.400	
Score LD	0.128	0.430	−0.097	0.551	−0.042	0.798	
Score ICC*	0.037	0.820	0.103	0.528	0.159	0.326	
Note:

SD, sensory domain; PD, psychological domain; CD, cognitive domain; VD, vitality domain; LD, locomotion domain; ICC, intrinsic capacity composite.

Discussion

This study is one of the few in Brazil that examines the effect of exercise interventions on the intrinsic capacity of community-dwelling older adults. Our findings demonstrate that a 12-week multicomponent training program significantly improved the composite intrinsic capacity score, particularly in the domains of cognition, vitality, and locomotion. Additionally, we observed that Chinese curative gymnastics alone positively influenced the composite intrinsic capacity score, as did combinations of strength training, gerontomotricity, and Chinese curative gymnastics.

Our results align with previous studies (Sánchez-Sánchez et al., 2022; Huang et al., 2021; Merchant et al., 2024; Ferrara et al., 2023), which have also reported improvements in composite intrinsic capacity scores following exercise interventions. The ICOPE guidelines already recommend multicomponent training for cognitive, vitality, and locomotion domains (World Health Organization, 2019). Moreover, our study extends these benefits to psychological and sensory domains, especially when exercises are performed in groups, which enhance social interaction and reduce loneliness and isolation.

These findings underscore the importance of exercise interventions in improving various aspects of intrinsic capacity in older adults, contributing to healthier aging and better quality of life. Physical exercise is a crucial lifestyle factor that aids in maintaining capacities and preventing functional decline, with benefits spanning various physiological systems. A study conducted with older adults in Mexico, a middle-income country like Brazil, showed that physically active participants were 1.7 times more likely to age healthily compared to those who were inactive (Arroyo-Quiroz, Brunauer & Alavez, 2020). Similarly, a study in France (Atallah et al., 2018) also demonstrated a positive association between exercise and healthy aging, corroborating the results of our study.

In the locomotor domain, we identified significant improvements in lower limb strength/power, functional mobility, flexibility, and balance in the post-test. Multicomponent interventions, including resistance exercises, are associated with increased muscle strength and power (Ferreira et al., 2012; Liu et al., 2014; Guizelini et al., 2018; Straight et al., 2016), leading to better performance in the sit-to-stand test (Ramsey et al., 2021). Exercise induces muscle damage that triggers an inflammatory response, recruiting immune cells to repair the injured area and increasing protein synthesis, which contributes to muscle fiber hypertrophy (Vandervoort, 2002). Larger muscle fibers can generate more force, thus enhancing muscle strength. Additionally, training induces adaptations in the nervous system, improving muscle activation efficiency, and leading to better coordinated and stronger muscle contractions (Schumann, Bloch & Oberste, 2019).

The time taken to complete the Timed Up and Go test decreased significantly in our sample, although participants already demonstrated good baseline performance. Previous studies on multicomponent training with community-dwelling older adults have shown similar results (Rodrigues et al., 2023; Nor et al., 2021; Yamada et al., 2013). The combination of exercise modalities presents a greater challenge to the musculature, reinforcing adaptive responses across various physical capacities (Rodrigues et al., 2022), thereby improving functional mobility and the ability to perform daily activities independently (Groessl et al., 2019).

Participants’ flexibility also improved significantly after 12 weeks of multicomponent training, supporting existing literature on the subject (Yan et al., 2023). Our program included flexibility-focused modalities such as Pilates and Chinese curative gymnastics. Stretching sessions were part of the protocol in other modalities to initiate exercise. Flexibility is a crucial health indicator for maintaining joint range of motion, contributing to postural balance and increasing both explosive (power) and maximal strength (Sobrinho et al., 2021; Fjerstad et al., 2018; Bucht & Donath, 2019). Flexibility training reduces actin-myosin filament overlap, leading to greater recruitment of inactive muscle fibers as compensation, thus enhancing physical capacities (Sekendiz, Cuğ & Korkusuz, 2010).

Similarly, the time participants could maintain a one-legged stance increased, indicating improvements in balance within the sample. Balance, the ability to remain standing or coordinate body movements, requires well-functioning physical components such as agility (Deng et al., 2024). Like in our research, exercise has been shown to effectively improve balance and reduce fall risk in older adults (Lesinski et al., 2015; Kim et al., 2020). Our findings regarding the locomotor domain align with other studies that have also investigated the benefits of multicomponent exercise in this dimension (Coelho Junior et al., 2017; Tarazona-Santabalbina et al., 2016; Freiberger et al., 2012; Kang et al., 2015; Toto et al., 2012).

Vitality scores increased significantly after the multicomponent training intervention, consistent with findings from other studies on physical training (Sánchez-Sánchez et al., 2022; Rainero et al., 2021; Clare et al., 2015). Notably, post-test analyses revealed differences in BMI and self-reported weight loss. Evidence suggests that lifestyle modification interventions, including regular physical exercise, can achieve up to 10% weight loss in older adults (Waters, Ward & Villareal, 2013; Villareal et al., 2006a, 2006b; Frimel, Sinacore & Villareal, 2008; Villareal et al., 2011), leading to a decrease in BMI.

Conversely, we did not observe improvements in upper limb strength, measured by handgrip strength. Training programs that include resistance exercises typically promote muscle strength increases even in very older adults (Borde, Hortobágyi & Granacher, 2015; Straight et al., 2016). This finding may be explained by the shorter training periods (4–16 weeks), which may not significantly influence strength gains compared to longer durations (52–53 weeks) (Guizelini et al., 2018).

Additionally, we did not include a nutritional intervention in the program, and protein and supplement intake, although controversial, can influence strength gains with resistance training (Choi, Kim & Bae, 2021). A study conducted by Rondanelli et al. (2016) implemented an intervention combining exercise instruction with formulated nutritional supplements, yielding favorable results in handgrip strength for the intervention group, which supports our inconclusive findings in this parameter. Additional evidence with resistance training and protein supplementation also supports beneficial outcomes for the intervention group (Seino et al., 2017; Mori & Tokuda, 2018; Nilsson et al., 2020; Zhu et al., 2019; Oesen et al., 2015).

In the cognitive domain, we observed an increase in MoCA test scores following participation in the program. The relationship between multicomponent exercise and cognitive benefits is well-documented (Tarazona-Santabalbina et al., 2016; Suzuki et al., 2013; De Asteasu et al., 2017; Forte et al., 2013), this is particularly evident when aerobic exercise is included in the intervention (Venegas-Sanabria et al., 2022). Herrero and colleagues investigated the effect of a multicomponent intervention using the Vivifrail program, a home-based exercise protocol with individual and personalized prescription. They found that the intervention group showed significant improvements in the MOCA test after 3 months of intervention (2.05 points; 95% CI [0.80–3.28]). In contrast, this trend was not observed in the control group (after 3 months, −0.13 points; 95% CI [−1.08 to 0.82]) (p < 0.05) (Casas‐Herrero et al., 2022).

Diverse exercises involve various neurobiological mechanisms, such as brain-derived neurotrophic factor, vascular endothelial growth factor, and inflammatory cytokines, which trigger biochemical responses favoring cognitive health (Liu et al., 2021; Morland et al., 2017; Tsai et al., 2019; Zaychik et al., 2021). Additionally, it has been suggested that resistance exercises increase lactate concentration in the bloodstream. Once in circulation, lactate reaches the brain and enhances the expression of genes associated with cognition (e.g., BDNF) (Huang et al., 2021; Yang et al., 2014).

The improvement may also be attributed to program modalities that provide intense neuromotor stimulation, combining cognitive tasks with exercise (Yan et al., 2023). Furthermore, exercises involving balance and motor coordination activate specific cortical-cerebellar connections that enhance cognitive function (De la Rosa et al., 2020). Meta-analyses conducted with healthy older adults have shown that studies implementing multicomponent interventions, including resistance, aerobic, and other exercises, were the most robust interventions for improving cognitive ability (Sanders et al., 2019; Kelly et al., 2014; Smith et al., 2010; van Uffelen et al., 2008; Colcombe & Kramer, 2003).

We did not observe significant improvements in the sensory domain, which includes vision and hearing. This was expected due to the complexity of these variables. The sensory system is influenced by various factors, such as natural aging, pre-existing medical conditions, and structural changes that cannot be easily modified through physical interventions, such as exercise (Saftari & Kwon, 2018; Sloane, Owsley & Jackson, 1988; Gates & Mills, 2005; Viljanen et al., 2009). Additionally, we did not include assistive technologies that could potentially influence these results. The absence of such limiting interventions may have contributed to the lack of observed improvements in the sensory domain, as they play a crucial role in compensating for sensory deficits in older adults.

Similarly, we did not observe significant differences in the psychological domain. However, individual analysis revealed that participants had an average score on the Geriatric Depression Scale classifying them as non-depressed, indicating a positive baseline condition in this specific aspect. This factor may have contributed to the lack of improvements in the psychological domain, as the lack of variability in the sample prevented the identification of significant changes. For clinically depressed older adults, physical exercise has proven antidepressant effects (Miller et al., 2020), which could have been more evident if there had been a higher prevalence of depressive symptoms in the initial group.

Chinese curative gymnastics was the only modality that alone positively influenced the composite intrinsic capacity score. This category of mind-body exercises includes various therapeutic practices such as Tai Chi, Qigong, Baduanjin, Yijinjing, and Lion Gong (the latter included in our study). These are characterized by gentle movements, breathing techniques, and low-to-moderate intensity exercises that can improve strength and overall health (Guo et al., 2016; Wang et al., 2016; Chen et al., 2015; Luo et al., 2016). They have broad adherence among older adults, especially those experiencing a decline in physical function and unable to tolerate high-intensity training (Bobbo et al., 2018).

Previous studies have demonstrated the benefits of mind-body exercises across various dimensions of intrinsic capacity, including improvements in global cognitive function (Yao et al., 2023; Cai et al., 2023) and increases in strength, balance, flexibility, and self-efficacy (Xu et al., 2018; Lolak et al., 2008). Additionally, these practices enhance balance control, proprioception, and postural adaptation, contributing to a reduced risk of falls in older adults (Tsang & Hui-Chan, 2004; Song et al., 2014; Huang et al., 2017; Jain et al., 2017).

These effects partially explain the results observed in our study, where Chinese curative gymnastics showed a significant improvement in the participants’ intrinsic capacity. It is worth noting that this modality was the most frequently chosen by older adults, likely due to its accessible and gentle nature, allowing participants to integrate physical and mental benefits easily. Higher adherence to this practice may have directly contributed to the gains in intrinsic capacity, reinforcing its potential as an effective intervention for promoting functional and mental health in the population.

Regarding strength training and gerontomotricity, these two modalities, when combined, act synergistically, amplifying the benefits to intrinsic capacity. Evidence from both exercise types indicates improvements in variables within the locomotor domain (Hayashi et al., 2021; Alqahtani et al., 2019), vitality (Labott et al., 2019; González-Rocha et al., 2022), cognition (Yoon, Lee & Song, 2018), and psychological outcomes (Chen et al., 2022), particularly when performed in groups. A systematic review highlighted an interactive relationship between the subdomains of intrinsic capacity, which explains why these two modalities have the potential for compensatory and interactive effects among components (Liao, Shen & Li, 2023).

Contrary to our hypothesis, weekly training frequency, training time, and the number of modalities showed no significant impact on intrinsic capacity scores. These findings suggest that training volume is not a crucial determinant. However, WHO recommendations suggest that older adults engage in at least 150 min of moderate or 75 min of vigorous physical activity per week (Bull et al., 2020), which aligns with the Physical Activity Guide of the Ministry of Health and the inclusion criteria of this study.

Likely, the quality of training, with supervision and guidance from trained professionals and the personalization and individualization of exercises according to each older adult’s capabilities, may have more influence on the composite intrinsic capacity score than volume alone. This hypothesis could be confirmed by future studies specifically analyzing training quality. Moreover, future research should explore the long-term effects of such interventions and their impact on other domains of intrinsic capacity, as well as the mechanisms underlying these benefits.

One potential limitation of this study is the non-randomized design and the use of a convenience sample, which may introduce selection bias and limit the generalizability of the findings. Additionally, the absence of randomization may compromise the ability to isolate intervention effects, increasing susceptibility to biases related to uncontrolled variables. However, this design was chosen due to the nature of the community-based program where the study was conducted, which provides a more realistic application of interventions in participants’ daily contexts.

While the personalized and tailored approach to exercise prescription is a strength, allowing for individualized interventions, it also creates variability in the implementation and adherence to the program. This heterogeneity in practice could influence the outcomes and make it challenging to attribute improvements solely to the multicomponent exercise program. Future studies could address this limitation by incorporating a randomized controlled trial design with a larger and more diverse sample, which would enhance the robustness and generalizability of the results.

Another limitation of the study is the relatively short duration of the intervention (12 weeks), which may not have been sufficient to capture the long-term effects of the multicomponent exercise program on intrinsic capacity. While significant improvements were observed in the cognitive, vitality, and locomotor domains, the study did not find significant changes in the sensory and psychological domains, possibly due to the limited intervention period. Additionally, considering the use of other measurement tools, such as specific scales for psychological domain evaluation and more objective tests for sensory domain assessment, would be important. Extending the intervention duration and incorporating follow-up assessments could provide a more comprehensive understanding of the sustained benefits and cumulative effects of the program across all intrinsic capacity domains.

Furthermore, future multicenter studies could evaluate the replicability of the results in different contexts and populations, providing a broader perspective on the effectiveness of the interventions. Analyzing specific subgroups within the studied population is also essential to identify which groups benefit most from the interventions, considering variables such as age, gender, socioeconomic status, and pre-existing health conditions.

The findings of this research offer a robust foundation for the development and implementation of health policies and programs aimed at promoting healthy aging. By leveraging existing resources and focusing on their reorganization, these strategies minimize the need for costly investments while maximizing the impact on older populations. In alignment with the Integrated Care for Older People (ICOPE) model, the study highlights the potential to reorient and integrate existing healthcare services to meet the specific needs of older adults. Additionally, the emphasis on community service engagement underscores key aspects for the success of such programs in fostering positive changes in health status.

Conclusions

The multicomponent exercise program significantly improved the intrinsic capacity of community-dwelling older adults over a 12-week period. Notable improvements were observed in cognitive function, vitality, and locomotion, highlighting the program’s effectiveness in promoting healthy aging. Personalized and supervised training, including modalities like Chinese curative gymnastics, strength training, and gerontomotricity, played a crucial role in these positive outcomes. These findings underscore the importance of tailored exercise interventions in enhancing various domains of intrinsic capacity, ultimately contributing to better quality of life and independence in older adults. Future research should explore the long-term benefits and mechanisms underlying these improvements.

Supplemental Information

Supplemental Information 1 Codebook.

Supplemental Information 2 Dataset.

Supplemental Information 3 Checklist.

Supplemental Information 4 Protocol for collecting physical variables.

Supplemental Information 5 Intervention details.

The Pontifical Catholic University of Rio Grande do Sul (PUCRS) provided access to the exercise program in which the data was collected. We used OpenAI’s ChatGPT to translate text from Portuguese to English.

Additional Information and Declarations

Competing Interests

Rafael Baptista is an Academic Editor for PeerJ.

Author Contributions

Sarah Giulia Felipe conceived and designed the experiments, performed the experiments, analyzed the data, prepared figures and/or tables, authored or reviewed drafts of the article, and approved the final draft.

Clarissa Biehl Printes performed the experiments, analyzed the data, authored or reviewed drafts of the article, exercise program and data collection, and approved the final draft.

Douglas Kazutoshi Sato conceived and designed the experiments, authored or reviewed drafts of the article, and approved the final draft.

Rafael Reimann Baptista conceived and designed the experiments, analyzed the data, authored or reviewed drafts of the article, and approved the final draft.

Human Ethics

The following information was supplied relating to ethical approvals (i.e., approving body and any reference numbers):

The study was approved by the Research Ethics Committee of Pontifical Catholic University of Rio Grande do Sul-PUCRS (approval number 5.517.315/CAAE: 60234322.1.0000.5336).

Data Availability

The following information was supplied regarding data availability:

The data is available at Zenodo: Bandeira, S. G. F. (2024). IMPACT OF A MULTICOMPONENT PHYSICAL EXERCISE PROGRAM ON INTRINSIC CAPACITY IN COMMUNITY-DWELLING OLDER ADULTS [Data set]. Zenodo. https://doi.org/10.5281/zenodo.13984498.

Bandeira Felipe, Sarah Giulia, 2023, “Effects of Multicomponent Training on the Intrinsic Capacity of Community-Dwelling Older Adults: Quasi-Experimental Study”, https://doi.org/10.7910/DVN/UI5SGL, Harvard Dataverse, V1.

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
