# Peer review of "Impact of a multicomponent physical exercise program on intrinsic capacity in community-dwelling older adults"

_PeerJ, doi:10.7717/peerj.19017_

## Round 0.1 · original submission · Major Revisions

Thank you for your submission. The reviewers have identified a number of concerns that must be addressed.

Reviewer 1 ·

Basic reporting

1. Clear and unambiguous, professional English used throughout. The manuscript is written in clear, professional English that meets academic standards. However, minor grammatical inconsistencies and typographical errors were noted. A thorough proofreading is recommended to enhance readability.
2. Literature references, sufficient field background/context provided. The introduction and background sections provide sufficient context, clearly outlining the relevance of intrinsic capacity and its domains to healthy aging. The cited references are up-to-date and relevant, supporting the research rationale. The manuscript could benefit from adding more international comparative studies to further contextualize the findings.
3. Professional article structure, figures, tables. Raw data shared. The article adheres to standard scientific structure, with well-organized sections. Figures and tables are relevant and appropriately labeled. Raw data has been provided and is consistent with the journal's data-sharing policy. However, Figure 1’s legend requires additional detail to improve interpretability.
4. Self-contained with relevant results to hypotheses. The manuscript is self-contained and addresses the stated hypotheses comprehensively. Results are well-aligned with the hypotheses, providing a coherent and focused presentation of findings.

Experimental design

1. Original primary research within Aims and Scope of the journal. The study is original and aligns with the journal's aims and scope, addressing the impact of multicomponent physical exercise on intrinsic capacity in older adults. The research question is novel and relevant to advancing knowledge in the field.
2. Research question well-defined, relevant & meaningful. The research question is clearly defined, addressing a significant gap in the literature on intrinsic capacity and exercise in older adults. The study’s objectives are explicitly stated and meaningful.
3. Rigorous investigation performed to a high technical & ethical standard. The investigation is conducted rigorously, adhering to ethical standards, as evidenced by the appropriate ethical approval and informed consent. However, the lack of randomization reduces the robustness of causal inferences.
4. Methods described with sufficient detail & information to replicate. The methods section is detailed, enabling replication. The exercise interventions and measurement tools are well-described. However, more details on how adherence was monitored and assessed would strengthen the study's replicability.

Validity of the findings

1. Impact and novelty not assessed. Meaningful replication encouraged where rationale & benefit to literature is clearly stated. The study’s findings contribute valuable insights to the literature, especially in the Brazilian context. While the study is not highly novel, it provides meaningful replication and extension of prior research. The rationale for replication is well-justified.
2. All underlying data have been provided; they are robust, statistically sound, & controlled. The data are robust, statistically sound, and adequately controlled. Statistical analyses, including Generalized Estimating Equations, are appropriate and well-documented. However, potential biases due to the non-randomized design should be more explicitly discussed.
3. Conclusions are well stated, linked to original research question & limited to supporting results. The conclusions are well-stated and appropriately linked to the research question. They are limited to the supporting results, highlighting improvements in cognition, vitality, and locomotion domains while acknowledging the lack of significant changes in sensory and psychological domains.

Additional comments

Recommendations for Improvement:
1. Explicitly discuss the potential limitations of the non-randomized design and implications for generalizability.
2. Enhance the discussion on how adherence to the exercise program was monitored and its impact on outcomes.
3. Provide additional detail in Figure 1’s legend to improve clarity.
4. Suggest future research directions, such as randomized controlled trials or longer-term interventions to assess sustained effects.

Reviewer 2 ·

Basic reporting

The study is very informative with regard to the contribution of multicomponent exercise in enhancing intrinsic capacity among elderly subjects. However, it suffers from methodological limitations regarding no randomization and an extremely short intervention period. Addressing these issues in future studies could substantially strengthen the evidence base for tailored exercise interventions in geriatric care.

The manuscript is well-written in clear, unambiguous, and professional English. The language is technically accurate, easy for an international audience to understand.
Some of those references, for example, references relating to some WHO frameworks could be supported with direct URL or more explicit citation formatting when it comes to accessibility, and methods require more precise detailing in some assessment tools and protocols, for example, self-reported assessments of sensory. This potentially jeopardizes reproducibility of the results. Raw data mentioned, needs to be sufficiently linked - in text - to the manuscript. Required is a direct appendix and/or repository link for raw data to be transparent and allow validations.

Experimental design

The design is quasi-experimental and non-randomized; therefore, it cannot allow for strong causal inferences. Though the design is representative of a natural setting, the absence of a control group diminishes the strength of the conclusions one may draw.
The convenience sampling method is subject to biases in selection and limits the generalization of results.
The training protocol should include details of further information, including progression criteria based on exercise intensity and specific supervision. Variability related to instructor-driven modalities is also noted. These should be controlled or systematically reported.
Though multi-modal exercise is commendable for its inclusion, the detailed study of the effects of its components, such as Chinese curative gymnastics versus resistance training, has to be done.

Validity of the findings

The authors do not adequately discuss the results in the sensory and psychological domains as not significant. A longer intervention period or other measurement tools should be discussed.
The conclusion exaggerates the generalizability of the results without appropriately considering limitations such as the short duration of the study and its non-randomized design.
There is a weak correlation analysis between the frequency of training and intrinsic capacity scores, hence limiting its explanatory value. Further studies should be carried out to examine the confounding factors.

Additional comments

Suggestions for Improvement

Major Revisions

An RCT design or at least the inclusion of a comparison group, for instance sedentary older adults, should be considered in order to strengthen the validity of the findings.
A longer intervention period would be more appropriate to capture its long-term effect on intrinsic capacity, especially for sensory and psychological domains.
Standardize training delivery across instructors and modalities to ensure consistency.
Fully explain how different exercise modalities, like Chinese curative gymnastics and resistance training, singly and in combination, affect the individual and combined responses.
Raw data should be openly available, either in a supplement or linked to via a repository.

Minor Revisions

Expand the discussion of null findings and alternative explanations.
What are the practical implications of this finding, if any, for policymakers and health providers?

---

## Round 0.2 · accepted · Accept

Thank you for your revised submission. I am satisfied that you have addressed the remaining concerns of the reviewers, and am happy to accept your paper for publication.

Reviewer 1 ·

Basic reporting

The author responded in detail to my concerns. It basically addressed my concerns about these issues.

Experimental design

no comment

Validity of the findings

no comment

Additional comments

no comment